# Circulating Exosomal miRNA Profiles Predict the Occurrence and Recurrence of Hepatocellular Carcinoma in Patients with Direct-Acting Antiviral-Induced Sustained Viral Response

**DOI:** 10.3390/biomedicines7040087

**Published:** 2019-11-03

**Authors:** Saori Itami-Matsumoto, Michiyo Hayakawa, Sawako Uchida-Kobayashi, Masaru Enomoto, Akihiro Tamori, Kazuyuki Mizuno, Hidenori Toyoda, Takeyuki Tamura, Tatsuya Akutsu, Takahiro Ochiya, Norifumi Kawada, Yoshiki Murakami

**Affiliations:** 1Department of Hepatology, Graduate School of Medicine, Osaka City University, 1-4-3 Asahimachi, Abeno-ku, Osaka 545-8585, Japan; sarinyan81@gmail.com (S.I.-M.); michi-hy@koto.kpu-m.ac.jp (M.H.); sawako@med.osaka-cu.ac.jp (S.U.-K.); enomoto-m@med.osaka-cu.ac.jp (M.E.); atamori@med.osaka-cu.ac.jp (A.T.); kawadanori@med.osaka-cu.ac.jp (N.K.); 2Department of Gastroenterology, Ogaki Municipal Hospital, 4-86 Minaminokawa-cho, Ogaki, Gifu 503-8502, Japan; kazu_miz075@yahoo.co.jp (K.M.); tkumada@he.mirai.ne.jp (H.T.); 3Bioinformatics Center, Institute for Chemical Research, Kyoto University, Gokasho, Uji, Kyoto 611-0011, Japan; tamura@kuicr.kyoto-u.ac.jp (T.T.); takutsu@kuicr.kyoto-u.ac.jp (T.A.); 4Institute of Medical Science, Tokyo Medical University, 6-7-1, Nishishinjuku, Shinjuku-ku, Tokyo 160-0023, Japan; tochiya@tokyo-med.ac.jp

**Keywords:** sustained viral response, direct-acting antiviral therapy, miRNA, exosome, microarray

## Abstract

Direct-acting antiviral (DAA) therapy for chronic hepatitis C virus (HCV) infection patients (CH) results in a sustained viral response (SVR) in over 95% of patients. However, hepatocellular carcinoma (HCC) occurs in 1–5% of patients who achieved an SVR after treatment with interferon. We attempted to develop a minimally invasive and highly reliable method of predicting the occurrence and recurrence of HCC in patients who achieved an SVR with DAA therapy. The exosomal miRNA expression patterns of 69 CH patients who underwent HCC curative treatment and 70 CH patients were assessed using microarray analysis. We identified a miRNA expression pattern characteristic of SVR-HCC by using machine learning. Twenty-five of 69 patients had HCC recurrence. The expression of four exosomal miRNAs predicted HCC recurrence with 85.3% accuracy. Fifteen of 70 patients had HCC occurrence. The expression of four exosomal miRNAs predicted the onset of HCC with 85.5% accuracy. The expression patterns of miR-4718, 642a-5p, 6826-3p, and 762 in exosomes were positively correlated with those in the liver, and downregulation of these miRNAs induced cell proliferation and prevented apoptosis in vitro. Aberrant expression of four miRNAs, which was used for prediction, was associated with HCC onset after HCV eradication. Expression patterns of exosomal miRNAs are a promising tool to predict SVR-HCC.

## 1. Introduction

Interferon (IFN) treatment of non-cirrhotic hepatitis C virus (HCV)-infected patients results in a sustained viral response (SVR) in approximately 50–60% of cases; however, 1–5% of these patients develop hepatocellular carcinoma (HCC) within 10 years of achieving an SVR [1]. Eliminating HCV and attaining an SVR is the goal of antiviral treatment because this reduces the risk of death by liver disease or of a decompensated state [2]. However, some patients develop HCC after achieving an SVR [3]; those with cirrhosis are particularly vulnerable. The risk factors for HCC after achieving an SVR include age, gender, liver fibrosis stage, and alpha-fetoprotein (AFP) level.

Direct-acting antiviral (DAA) treatment reduces the risk of HCC and results in an SVR in approximately 95–100% of patients. DAAs have mild side effects and the treatment is of a shorter duration than that of IFN treatment. The IFN treatment is typically geared toward younger patients and those with a relatively low stage of liver fibrosis; elderly patients and patients with advanced liver fibrosis are preferred candidates for DAA treatment. Over 95% of patients treated with DAAs achieve an SVR; however, the risk of HCC remains high after eliminating HCV. In two large-scale studies involving patients without HCC, a DAA-induced SVR was associated with a 71–76% reduction in the frequency of initial HCC occurrence [4,5]. Therefore, DAAs are recommended for patients who have not undergone curative treatment for HCC.

The frequency of HCC intrahepatic metastasis and/or recurrence after DAA treatment in patients with prior HCC is relatively high. After a median follow-up of 5.7 months, 16 of 58 DAA-treated patients developed HCC [6], with a recurrence rate of 29.1% [7]. The probability of early recurrence of HCC (within 1 year) in patients who had prior curative HCC treatment is high; this is also true for intrahepatic metastasis in patients in whom DAAs eradicated HCV. After achieving an SVR, patients with HCC should be monitored twice a year indefinitely via liver imaging and AFP testing.

Exosomes are small membranous vesicles that are implicated in viral infection. They also promote cellular immune responses via cell-cell communication. Exosomes contain microRNAs (miRNAs) and may be involved in immune-independent regulatory mechanisms. Exosomes mediate intercellular transfer of miRNAs. As such, they participate in miRNA-based signaling [8]. HCV infection induces HCC via a combination of pathway alterations that are caused by viral replication or chronic inflammation [9,10]. Exosomes are also implicated in propagating HCV RNA between hepatocytes [11]. Exosomal miRNA expression patterns enable the diagnosis of not only chronic liver disease but also the stage and grade of liver injury [12]. This is achieved by selecting the circulating miRNAs appropriate for use as disease biomarkers [13,14]. Diagnosis of HCC in its early stages influences the therapeutic approach and the prognosis. Biomarker measurements are crucial for early detection, monitoring, and evaluation of the treatment response [15]. Exosome-based biomarkers have potential utility for HCC [16,17].

We investigated the association between SVR-HCC and exosomal miRNA expression patterns in patients with and without liver cirrhosis 12 weeks after achieving an SVR (SVR12). In this study, we fractionated miRNA in blood with ExoQuick. Although exosome extraction by ExoQuick is simpler than the ultracentrifugation method, which is the gold standard, it is known to contain RNA-binding proteins in addition to vesicles. Therefore, our method was to purify circulating miRNA with ExoQuick (hereafter, exosomal miRNA) [18,19]. We also developed an accurate and minimally invasive method of predicting initial HCC occurrence and its recurrence in patients with a history of curative treatment for HCC.

## 2. Experimental Section

### 2.1. Sample Information

Serum was obtained from 139 patients 12 weeks after achieving a DAA-induced SVR (SVR12). Sixty-two patients were treated with asunaprevir and daclatasvir, 63 with ledipasvir and sofosbuvir, and the remaining 14 were administered sofosbuvir and ribavirin. A total of 69 patients had a history of curative HCC treatment and 70 had not had HCC prior to undergoing DAA treatment at Osaka City University Hospital. Liver tissue was obtained from 43 patients before DAA treatment at Ogaki Municipal Hospital (asunaprevir and daclatasvir, 28 patients; ledipasvir and sofosbuvir, six patients; sofosbuvir and ribavirin, four patients; and ombitasvir, paritaprevir, and ritonavir, five patients) (Figure 1, Table 1 and Appendix A). This study was conducted according to the guidelines of the 1975 Declaration of Helsinki (2013 version). Written informed consent was obtained from all patients prior to treatment. The study protocol was approved by the Ethics Committees of Osaka City University Hospital (Project code No. 1358, 12 June 2008) and Ogaki Municipal Hospital (Project code No. G-219, 7 May 2007) respectively.

### 2.2. RNA Preparation and Microarray Assay

Exosome-rich fractionated RNA was prepared using ExoQuick (System Biosciences, Palo Alto, CA, USA). RNA was extracted from exosomes and liver tissue using a miRNeasy Mini Kit (Qiagen, Hilden, Germany).

Sixty nanograms of total RNA were analyzed using the 3D-Gene miRNA microarray RNA extraction reagent from the liquid sample kit (Toray Industries, Inc., Kanagawa, Japan). A comprehensive miRNA expression analysis was performed using a 3D-Gene miRNA Labeling Kit and a 3D-Gene Human miRNA Oligo Chip (Toray Industries, Inc), both of which could detect 2555 miRNA sequences in miRBase release 20 (http://www.mirbase.org/). All microarray data for this study conformed to the Minimum Information about a Microarray Experiment guidelines and are publicly available in the GEO database (GSE119156 for liver tissue and GSE119159 for exosomes).

### 2.3. Cell Proliferation Assay

Cell proliferation assays were performed using the XTT Cell Proliferation Assay Kit (Roche, Basel, Switzerland). Briefly, Huh7.5 (5.0 × 10^3^/well) and HepG2 (1.0 × 10^4^/well) cells were spread into 96-well dishes. Four picomoles of annealed double stranded miRNA: mature miRNA and short RNA of which the sequence was complementary, miR-4718, 6511a-5p, 642a-5p, 4448, 211-3p, 6826-3p, 1236-3p, 762, and 8069, and negative control siRNA (Silencer Negative Control No.1 siRNA; Thermo Fisher Scientific, Waltham, MA, USA) were transfected using Lipofectamine RNAiMAX (Invitrogen, Carlsbad, CA, USA). After 24 and 48 h, 50 µL of XTT labeling mixture was added and the cells were incubated in a humidified atmosphere for 6 h at 37 °C. Next, the absorbance at 450–500 nm was measured using an enzyme-linked immunosorbent assay (ELISA) reader with a reference wavelength of 650 nm.

### 2.4. Caspase Assay

Caspase-9 assays were performed using the Caspase-Glo 9R Assay Kit (Promega, Madison, WI, USA). Briefly, huh7.5 (5.0 × 10^3^/well) and HepG2 (1.0 × 10^4^/well) cells were spread into 96-well light-shaded dishes. After 6 h, the cells were incubated for 15 min at room temperature, caspase-9 substrate mixture was added, and the cells were incubated for 15 min at room temperature. Next, the luminescence was measured and caspase-9 activity was normalized to the amount of XTT incorporated.

### 2.5. HCC Prediction by Linear SVM

To select miRNAs that enabled discrimination of SVR-HCC and SVR-non-HCC, we used a simple greedy algorithm using a linear-SVM (Support Vector Machine). The SVM is a machine learning method for the classification of samples into positive (e.g., cancer samples) and negative ones (e.g., normal samples). The SVM has two phases: learning and prediction. The SVM uses training samples in the learning phase to derive a discrimination rule, which is represented by a hyperplane (i.e., linear equality) in the case of a linear-SVM. In the prediction phase, the SVM classifies each new sample as positive or negative. SVMs have successfully classified various diseases. For the details of SVMs and their applications in medical problems, see the review article by Huang et al. [20]. In this study, we mainly employed a linear-SVM that uses a linear kernel because it is considered to be more robust to overfitting than other kernels, and interpretation of the classification rules is easier than in other models.

The selection procedure was as follows:

Exclude miRNAs with very low expression (log expression score ≤1.0, which can be regarded as a measurement error) in 10% or more of the samples.

Select the top 50 miRNAs based on their *F*-value determined by analysis of variance (ANOVA).

Perform a cross-validation test on pairs of the 50 selected miRNAs, and select the pair with the highest accuracy.

Repeat (3) above until the specified number of miRNAs is reached. Perform a cross-validation test on each of the remaining miRNAs by temporarily adding it to the current set; finally, choose the best miRNA and add it to the current set.

In all computational experiments, we adopted the svm.SVC function in the scikit_learn machine learning toolbox in Python 3.6 on a Linux/Cygwin operating system using C = 1,000, kernel = linear, and class_weight = balanced options. To select 50 miRNAs using the ANOVA-determined F value, we adopted the f_classif function of scikit_learn. In each step, cross-validation was performed 100 times using 70% of the data for training and 30% for testing, and the average score was used. Note that the set of miRNAs selected can vary from test to test because of the intrinsic randomness of the cross-validation procedure. Therefore, for each case, we determined the set of features based on the results of 20 executions.

To assess the reliability of the predictions, 1000 cross-validations were performed using 70% of the data for training and 30% for testing. We calculated the average accuracy (ACC), recall (REC), precision (PRE), and specificity (SPE) for over 1000 trials, defined as follows: (1)$$\mathrm{accuracy}=\frac{\mathrm{TP}+\mathrm{TN}}{\mathrm{TP}+\mathrm{FP}+\mathrm{TN}+\mathrm{FP}},$$
(2)$$\mathrm{recall}=\frac{\mathrm{TP}}{\mathrm{TP}+\mathrm{FN}}$$
(3)$$\mathrm{precision}=\frac{\mathrm{TP}}{\mathrm{TP}+\mathrm{FP}},$$
(4)$$\mathrm{specificity}=\frac{\mathrm{TN}}{\mathrm{TN}+\mathrm{FP}}$$ where TP, TN, FP, and FN represent the number of correctly predicted positive samples, correctly predicted negative samples, incorrectly predicted positive samples and incorrectly predicted negative samples, respectively. Note that sensitivity is equivalent to recall. Patients with HCC recurrence were regarded as positive. In addition, we calculated accuracy by leave-one-out cross-validation (LOOCV) and the accuracy when all samples were included in both the training and test datasets (ACCALL). Because the discrimination rule was always uniquely determined in the case of ACCALL, the discrimination rule and the corresponding classification results were also calculated. In addition, Receiver Operating Characteristic (ROC) curves were drawn where all samples were included in both the training and test datasets (Appendix A). Note that because the number of positive samples was very small, we could draw ROC curves only for such cases. To visualize the distribution of miRNA expression, heat maps were also created by applying supervised clustering for samples where 50 relevant miRNAs selected by the procedure in Section 2.5 were used for each case (Appendix A).

Because overfitting can occur in the above procedure, we performed a permutation test. Positive and negative samples were randomly switched to select four miRNAs, and the average and maximum accuracy were calculated for over 20 trials of the miRNA selection procedure.

### 2.6. Statistics

Data were analyzed using Student’s t-test, and *p*-values < 0.05 were considered significant.

## 3. Results

### 3.1. Predicting HCC Recurrence in Patients with DAA-Induced SVR Liver Cirrhosis

A group of 41 cirrhosis patients with a history of HCC treatment was identified as being at high risk of HCC recurrence. Prior to DAA treatment, the patients were confirmed to be free of HCC based on their blood AFP levels and results of dynamic liver computed tomography. After achieving SVR12, there were 16 cases of recurrent HCC (SVR-HCC) and 25 non-recurrent cases (SVR-non-HCC) (Figure 1A).

HCC emerged from 22 to 503 days post-SVR12 (mean, 243 days). Clinically, the serum albumin level before DAA treatment in SVR-non-HCC patients was significantly higher than that in SVR-HCC patients (*p* < 0.05) (Table 1 and Appendix A). We assessed liver fibrosis using a mac-2 binding protein glycosylation isomer (M2BPGi), a novel marker of the progression of liver fibrosis in patients with chronic HCV infection [21], and the FIB-4 index [22].

The exosomal miRNA expression pattern in SVR12 patients was evaluated using microarray analysis. The expression levels of 14 miRNAs in SVR-non-HCC patients were significantly lower than those in SVR-HCC patients; moreover, the expression of 42 miRNAs was significantly higher in SVR-non-HCC than in SVR-HCC patients (*p* < 0.05) (Appendix A). HCC occurring within 1 year post-treatment could be intrahepatic metastasis or recurrence, but because of the small sample size, both instances were treated as recurrence in this study.

For this group of patients with liver cirrhosis, we chose three miRNAs (miR-4718, 6511a-5p, and 642a-5p) using the linear-SVM-based selection procedure described in Section 2.5. The resulting discrimination rule was as follows: y = −4.12 × [miR-4718] + 3.46 × [miR-6511a-5p] − 2.08 × [miR-642a-5p] + 13.51(5)

Note that this equation represents a hyperplane obtained as a result of the learning phase of a linear-SVM. In the prediction phase, y < 0 indicated SVR-non-HCC and y ≥ 0 indicated SVR-HCC. This case performed as follows: ACC = 88.5%, REC = 89.5%, PRE = 83.0%, SPE = 87.8%, LOOCV = 92.7%, and ACCALL = 97.6%. The predictions obtained by this rule are given in Table 2.

LOOCV = 92.7%, and ACCALL = 97.6%. The predictions obtained by this rule are given in Table 2. The average and maximum accuracies of the permutation tests were 70.1% and 85.2%, respectively, and only one of the 20 cases exceeded 80% accuracy. Thus, the likelihood of overfitting was low.

Four miRNAs (miR-4718, 6511a-5p, 642a-5p, and 4448) enabled SVR-non-HCC and SVR-HCC to be distinguished with 87.2% accuracy, compared to the 79.6% accuracy using only miR-4718 and -6511a-5p. The prediction is shown in Figure 2 and Appendix A. Predictions made using three or four miRNAs were more accurate than those based on only two miRNAs. The above results suggest that these three miRNAs were highly predictive of SVR-HCC in patients at a high risk of cirrhosis.

### 3.2. Predicting HCC Recurrence in DAA-Induced SVR Liver Cirrhosis and Non-Cirrhosis Patients

After achieving SVR12, 25 of 69 patients had recurrent HCC (Figure 1B). HCC recurrence appeared from 22 to 588 days after SVR12 (mean, 235 days). The albumin level before DAA treatment and the AFP level after DAA treatment differed significantly between patients with and without recurrent HCC (Table 3 and Appendix A).

The expression levels of 34 and 36 miRNAs were significantly lower and higher, respectively, in SVR-non-HCC patients than in SVR-HCC patients (*p* < 0.05) (Appendix A). For this prediction, we selected four miRNAs (miR-211-3p, 6826-3p, 1236-3p, and 4448) by linear-SVM. y = 1.62 × [miR-211-3p] − 1.43 × [miR-6826-3p] + 0.65 × [miR-1236-3p] − 0.84 × [miR-4448] − 1.19(6)

The resulting performance and the discrimination rule were: ACC = 85.3%, REC = 86.1%, PRE = 77.8%, SPE = 85.3%, LOOCV = 87.0%, and ACCALL = 87.0%. The prediction based on this rule is given in Table 4.

The mean and maximum accuracies of the permutation test were 69.0% and 75.2%, respectively. This result suggests that the possibility of overfitting was low.

The accuracy of discrimination using three miRNAs (miR-211-3p, 6826-3p, and 1236-3p) was 82.9%, and that of using two miRNAs (miR-211-3p and 6826-3p) was 79.0%. The prediction based on this rule is shown in Figure 2B and Appendix A. Similar to the patients at high risk of cirrhosis, the use of two miRNAs to make a prediction did not produce satisfactory results. Thus, our hypothesis is that two miRNAs are insufficient for highly accurate predictions in high- and low-risk cases.

We next attempted to predict SVR-HCC recurrence using a single miRNA. In patients with liver cirrhosis, the prediction accuracy was from 0.60 to 0.76 when each of the 10 miRNAs was individually used. In both non-cirrhosis and liver cirrhosis patients, the prediction accuracy was from 0.60 to 0.73 when each of 11 miRNAs was used (Appendix A).

### 3.3. Predicting HCC in HCC-Naïve Patients with a DAA-Induced SVR

After achieving SVR12, 15 patients with no history of HCC developed HCC prior to June 2, 2017 (SVR-HCC), and 55 patients did not (SVR-non-HCC) (Figure 1C). HCC was detected after 0 to 773 days (mean, 266 days). The M2BPGi level before DAA treatment and the AFP level after DAA treatment differed significantly between HCC naïve patients that developed HCC and those that did not (*p* < 0.05) (Table 5 and Appendix A). The expression levels of 118 and 258 miRNAs were significantly lower and higher, respectively, in SVR-non-HCC than in SVR-HCC patients (*p* < 0.05) (Appendix A).

For this dataset, the prediction using linear-SVM-based selection was not highly accurate. Therefore, we used the radial basis function (RBF) kernel-based selection procedure with normalized gene expression data. Normalization was performed independently for each cell after step 1 (i.e., exclusion of low-expression genes), such that the resulting expression values had a mean of 0.0 and variance of 1.0. For the RBF kernel, we adopted the svm.SVC function in scikit_learn using the C = 1000, kernel = rbf, and class_weight = balanced options with the default gamma value.

Two miRNAs (miR-762 and 8069) were selected by the RBF kernel-based procedure. The resulting performance and the discrimination rule were: ACC = 83.3%, REC = 71.4%, PRE = 60.6%, SPE = 87.1%, LOOCV = 85.7%, and ACCALL = 85.7%. Because the RBF kernel was used, discrimination rules cannot be expressed in a simple form. The mean and maximum accuracies of the permutation test were 77.1% and 82.2%, respectively. Because the maximum accuracy is similar to that for the original (i.e., non-permutated) dataset, there is a risk of overfitting for this dataset. Overfitting is almost inevitable in this case because there are fewer positive samples (SVR-HCC) than negative samples. The low recall and precision values may also be caused by this imbalance. The ACCs of predictions using three and four miRNAs were 83.2% (miR-762, 8069, and 6090) and 85.2% (miR-762, 8069, 7847-3p, and 7846-3p) (Figure 3, Table 6, and Appendix A). However, the maximum accuracies of permutation tests were 86.4% and 86.7%, respectively. Therefore, the use of three or four miRNAs did not yield accurate results. Note also that area under the curve (AUC)-score of the ROC curve for two miRNAs was worse than those for three miRNAs and four miRNAs (Appendix A). However, this was also considered to be a result of overfitting for the cases of three and four miRNAs.

Vertical axis, miRNA relative expression level and class (1, no recurrence; 2, recurrence); horizontal axis, case. Upper panel: Relative miRNA expression level. Lower panels: Vertical axis, the predictive value calculated by the indicated equation (Table 1). Horizontal line (value = 0), threshold. Dots, predictive values were yielded individually by the indicated equation (Table 1). Four fractions divided by the threshold value and recurring HCC (yellow) and non-recurring HCC (white); lower left and lower right fractions show expected results, and upper left and lower right fractions show unexpected results.

### 3.4. Hepatic miRNA Expression Pattern in SVR-non-HCC and SVR-HCC Patients

We analyzed the miRNA expression patterns in 43 liver biopsy specimens obtained before DAA treatment from HCC-naïve patients. Seven of these forty-three patients developed HCC prior to January 24, 2017. The alanine transaminase (ALT), platelet, albumin, and AFP levels before DAA treatment differed significantly between the SVR-HCC and SVR-non-HCC patients (Table 7 and Appendix A).

According to microarray analysis, the expression levels of 323 and 95 miRNAs were significantly higher and lower, respectively, in SVR-HCC patients than in SVR-non-HCC patients (*p* < 0.05) (Appendix A).

### 3.5. Hepatocarcinogenic Potential of the miRNAs Used to Predict SVR-HCC

To analyze the function of nine miRNAs (miR-4718, 6511a-5p, 642a-5p, 4448, 211-3p, 6826-3p, 1236-3p, 762, and 8069), we assayed the proliferation and apoptosis of Huh7.5 and HepG2 cells.

Overexpression of miR-642a-5p, 6511a, and 211-3p at 24 h in both cell lines, and miR-8069 at 48 h in both cell lines inhibited cell proliferation (*p* < 0.05) (Figure 4A). Overexpression of miR-4718 and 642a-5p induced apoptosis in both cell lines (*p* < 0.05) (Figure 4B).

To assess the biological roles of the exosomal miRNAs, we investigated their expression patterns in exosomes and in the liver. We identified that four (miR-4718, 642a-5p, 6826-3p, and 762) of nine miRNAs had similar expression patterns in exosomes and the liver. ADP-ribosylation factor 6 (ARF6) [23], heat shock 70 kDa protein 1A (HSPA1), heat shock 70 kDa protein 1B (HSPA1B), low-density lipoprotein receptor (LDLR) [24], and Ras-related protein Rab-5B (RAB5B) [25] were regulated by miR-642a-5p. miRNA target genes that have a common expression in liver tissue and exosome were chosen (Figure 5).

Comparison of exosome and liver miRNA expression level by microarray was performed. Comparison of cases with (R+) or without (R-) relapse of HCC in exosome study and cases with (HCC+) or without (HCC-) HCC in liver tissue were combined. Broad arrowheads indicated the biological function, which demonstrated in this study. Narrow arrowheads indicated the target genes of miR-642a-5p in previously reports.

## 4. Discussion

We investigated the association between the expression patterns of exosomal miRNAs and the outcome of DAA treatment in HCV patients who achieved an SVR. The clinical information of each group was homogenized in order to obtain better prediction method; (1) HCC recurrence in the presence of advanced fibrosis, (2) HCC recurrence, and (3) first HCC occurrence. We achieved accurate SVR-HCC predictions by using fewer miRNAs than in the other groups. Although different algorithms were used for the first HCC occurrence and HCC recurrence groups, accurate prediction of the onset of HCC was possible.

To show the biological significance of miRNA used for hepatocarcinogenesis prediction, the target genes of miRNAs were clarified. The point we noted was that miRNA expression information of exosomes was used for prediction; however, miRNA did not control the target genes in the exosomes. We identified common expression patterns of HCC-related miRNAs in exosomes and liver tissue. Four (miR-4718, 642a-5p, 6826-3p, and 762) of nine miRNAs had similar expression patterns in exosomes and the liver. Downregulation of miR-642a-5p induced cell proliferation and down-regulation of miR-4718 and 642a-5p inhibited apoptosis in vitro. Down-regulation of miR-4718, 6823-3p, and 762 showed cell growth by observing one point of one cell line, respectively. Down-regulation of miR-6826-5p and miR-762 showed apoptosis induction in Huh7.5 in vitro. The downregulation of miR-642a-5p: (1) induced cell invasion by altering the expression of ARF6 [23]; (2) promoted tumor growth by influencing the expression of HSPA1, HSPA1B, and LDLR [24]; and (3) promoted cell proliferation by modulating the expression of RAB5B [25]. Thus, multiple miRNAs play synergistic roles in carcinogenesis (Figure 5).

Treatment for HCV is progressing; for example, IFN-free DAA treatment, which has fewer contraindications and side effects, has been introduced. However, approximately 1–5% of patients develop HCC, which can recur after achieving an SVR. Our results strongly suggest that HCC can be predicted with high accuracy using the expression patterns of exosomal miRNAs in patients who have achieved DAA-induced SVR. This analysis has some limitations, such as the observation period and the number of observation cases. We are continuing to follow-up, increasing the number of cases, and updating the results. There is also the possibility of overfitting when predicting HCC using our technique. Our results will facilitate follow-up studies of the DAA-induced SVR and of the mechanism for HCC in the absence of hepatitis viral infection.

## Figures and Tables

**Figure 1 biomedicines-07-00087-f001:**
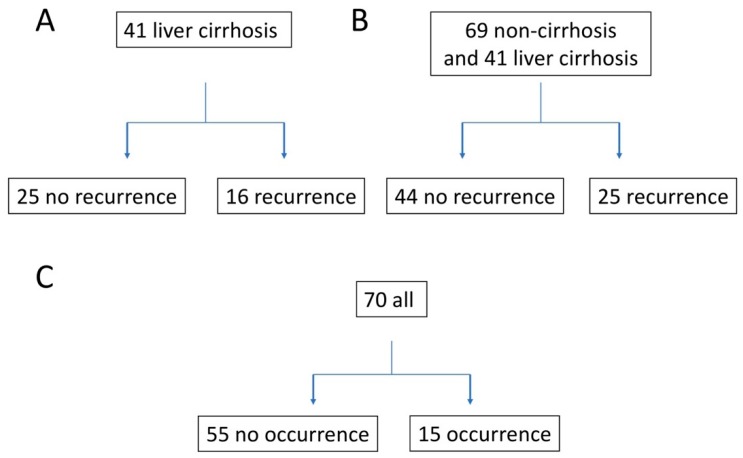
Experimental design. (**A**). Composition of the number for predicting hepatocellular carcinoma (HCC) recurrence in patients with Direct-acting antiviral (DAA)-induced SVR liver cirrhosis by exosome study: After treating cirrhosis in 41 patients with a history of liver cancer treatment, 25 patients had no cancer recurrence and 16 had cancer recurrence. (**B**). Composition of the number for predicting HCC recurrence in DAA-induced sustained viral response (SVR) liver cirrhosis and non-cirrhosis patients by exosome study: After treating DAA in 69 patients with chronic hepatitis and cirrhosis who had a history of liver cancer treatment, 44 had no recurrence of cancer and 25 had a recurrence of cancer. (**C**). Composition of the number for predicting HCC in HCC-naïve patients with a DAA-induced SVR by exosome study: After DAA treatment in 70 patients with chronic hepatitis and cirrhosis who had no history of liver cancer treatment, 55 did not develop cancer, and 15 developed cancer.

**Figure 2 biomedicines-07-00087-f002:**
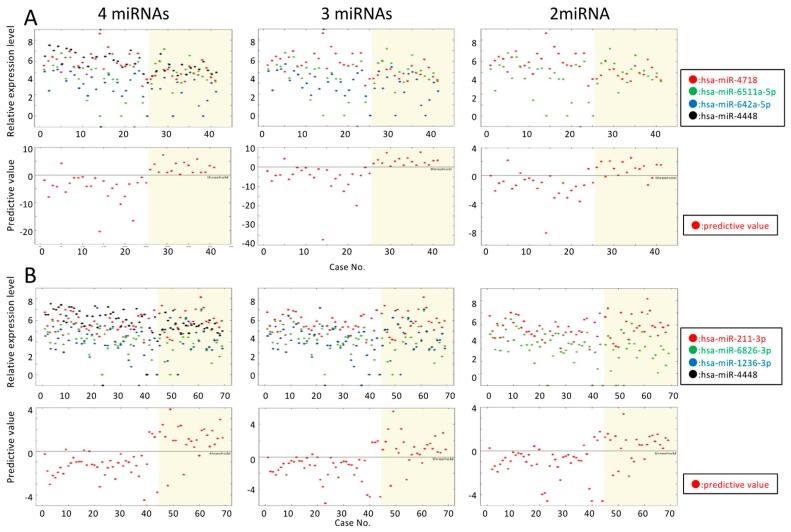
Predicting hepatocellular carcinoma (HCC) recurrence. (**A**). Prediction of HCC recurrence in patients with liver cirrhosis using 4, 3, and 2 miRNAs, respectively. (**B**). Prediction of HCC recurrence in non-cirrhosis and liver cirrhosis patients using 4, 3, and 2 miRNAs, respectively.

**Figure 3 biomedicines-07-00087-f003:**
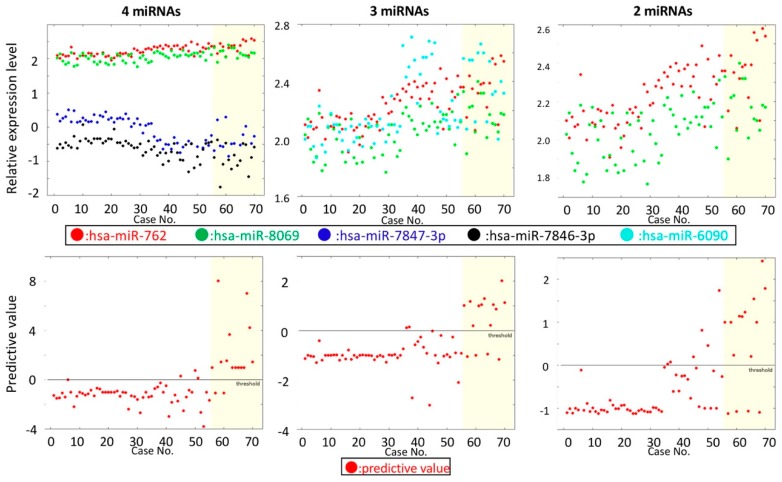
Predicting hepatocellular carcinoma (HCC) occurrence.

**Figure 4 biomedicines-07-00087-f004:**
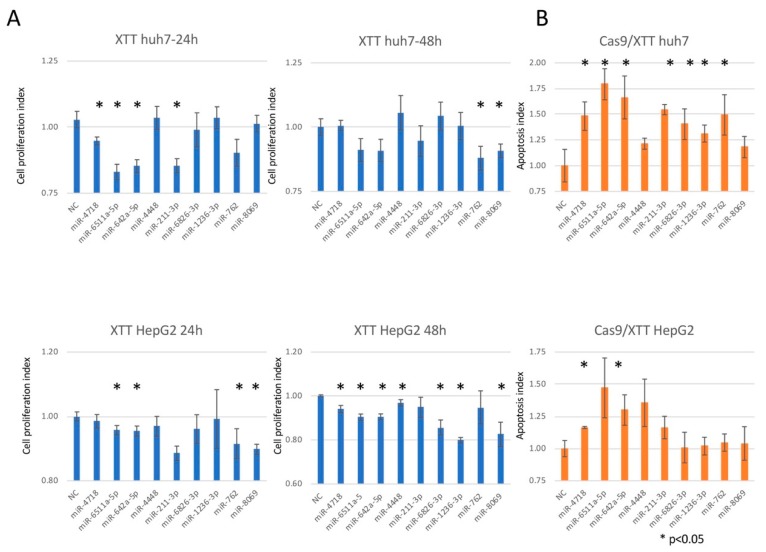
Functions of the miRNAs in cell proliferation and apoptosis. (**A**). Cell proliferation index in Huh7.5 and HepG2 cells 24 and 48 h after miRNA transfection. NT and TF, non-treatment and transfection reagent only, respectively. (**B**). Apoptosis assay by caspase-9 activity normalized to XTT incorporation in Huh7.5 and HepG2 cells 6 h after miRNA transfection. Data are the means ± SD of three independent experiments. Asterisks denote a significant difference (*p* < 0.05).

**Figure 5 biomedicines-07-00087-f005:**
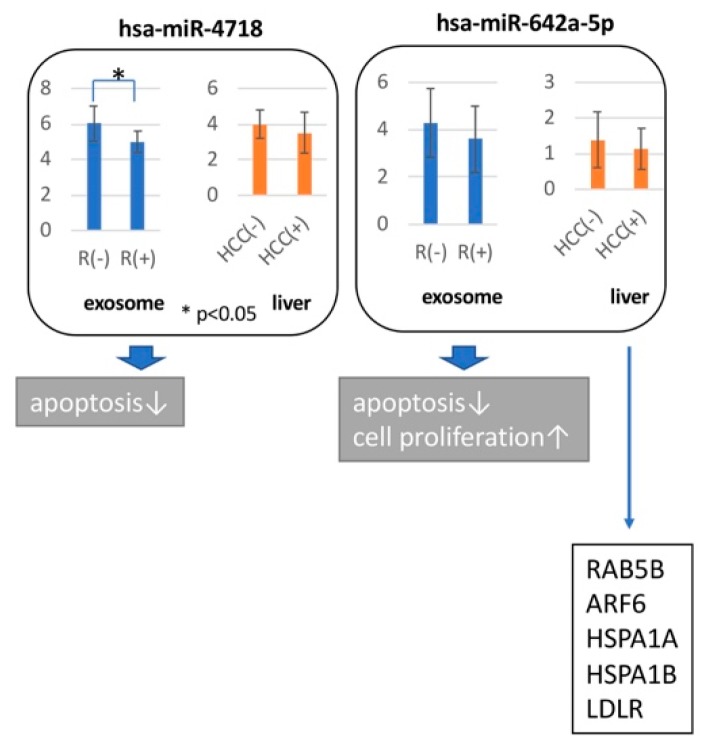
Hepatocarcinogenesis pathways involving miRNAs predictive of HCC.

**Table 1 biomedicines-07-00087-t001:** Clinical background. DAA treatment after HCC treatment in liver cirrhosis cases (exosome study).

Summary of Analysis Groups					
Presence or Absence of HCC Recurrence	No.	Age	Sex	Liver Disease	DAA Treatment
No	25	73.6 ± 7.2	F:13/M:12	liver cirrhosis	AD:8/SL:15/SR:2
Yes	16	72.1 ± 9.0	F:8/M:8	liver cirrhosis	AD:8/SL:4/SR:4
**Clinical Data by Analysis Group**					
**Treatment**	**Items**	**No Recurrence**	**Recurrence**	***p*-Value**	
pre-	ALT	46.8 ± 23.3	55.8 ± 31.1	3.02 × 10^−1^	
	T.Bil	0.8 ± 0.4	0.8 ± 0.3	6.67 × 10^−1^	
	**Alb**	**3.7 ± 0.3**	**3.3 ± 0.9**	**5.27 × 10^−3^**	
	PT	82.6 ± 8.1	75.9 ± 19.8	1.33 × 10^−1^	
	PLT	9.9 ± 2.9	9.1 ± 4.1	4.69 × 10^−1^	
	AFP	27.3 ± 58.7	42.5 ± 68.3	4.63 × 10^−1^	
	M2BPGi	6.0 ± 3.9	7.6 ± 3.9	2.24 × 10^−1^	
	FIB-4	6.6 ± 2.6	8.3 ± 4.3	1.27 × 10^−1^	
post-	ALT	44.6 ± 106.3	26.3 ± 15.8	5.01 × 10^−1^	
	T.Bil	0.9 ± 0.4	0.9 ± 0.3	4.17 × 10^−1^	
	Alb	3.8 ± 0.3	3.8 ± 0.4	7.78 × 10^−1^	
	PT	80.4 ± 19.2	78.1 ± 15.7	5.85 × 10^−1^	
	PLT	11.0 ± 3.5	9.8 ± 33.6	3.60 × 10^−1^	
	AFP	8.6 ± 7.2	14.5 ± 14.9	1.37 × 10^−1^	
	M2BPGi	3.8 ± 2.6	4.7 ± 3.1	3.00 × 10^−1^	

Abbreviations: AD, asunaprevir and daclatasvir; SL, ledipasvir and sofosbuvir; SR, sofosbuvir and ribavirin; VIK, ombitasvir, paritaprevir, and ritonavir; pre, sampling before DAA treatment; post, sampling; fibrosis stage, fibrosis was determined by histological study before DAA treatment. Bold indicates a significant difference between the two groups (*p* < 0.05).

**Table 2 biomedicines-07-00087-t002:** Summary of the equation and prediction score for recurring HCC in liver cirrhosis cases.

#miRNAs	Equation	ACC (%)	REC (%)	PRE (%)	SPE (%)	LOOCV (%)	ACCALL (%)
4	y = −2.75 × [miR-4718] + 3.02 × [miR-6511a-5p] −1.38 × [miR-642a-5p] −1.21 × [miR-4448]+12.60	87.2	87.8	83.1		92.7	97.6
**3**	**y = −4.12 × [miR-4718] + 3.46 × [miR-6511a-5p] − 2.08 × [miR-642a-5p] + 13.51**	**88.5**	**89.5**	**83.0**	**87.8**	**92.7**	**97.6**
2	y = −1.72 × [miR-4718] + 0.74 × [miR-6511a-5p] + 5.83	79.6	77.0	73.6		80.5	82.9

Abbreviations: Bold indicates Equation (5).

**Table 3 biomedicines-07-00087-t003:** Clinical background of DAA treatment after HCC treatment in all cases (exosome study).

Summary of Analysis Groups					
Presence or Absence of HCC Recurrence	No.	Age	Sex	Liver Disease	DAA Treatment
No	44	73.5 ± 7.76	F:20/M:24	non-cirrhosis:19 /liver cirrhosis:25	AD:15/SL:26/SR:3
Yes	25	72.7 ± 8.20	F:11/M:14	non-cirrhosis:9 /liver cirrhosis:16	AD:10/SL:9/SR:6
**Clinical Data by Analysis Group**					
**Treatment**	**Items**	**No Recurrence**	**Recurrence**	***p*-Value**	
pre-	ALT	49.1 ± 33.5	53.4 ± 35.4	6.13 × 10^−1^	
	T.Bil	0.7 ± 0.3	0.8 ± 0.3	8.76 × 10^−1^	
	**Alb**	**3.7 ± 0.3**	**3.5 ± 0.8**	**3.61 × 10^−2^**	
	PT	86.7 ± 13.4	81.2 ± 19.3	1.73 × 10^−1^	
	PLT	13.2 ± 6.1	10.8 ± 4.6	9.26 × 10^−2^	
	AFP	19.6 ± 45.1	39.6 ± 71.4	1.75 × 10^−1^	
	M2BPGi	4.5 ± 3.7	5.9 ± 4.1	1.44 × 10^−1^	
	**FIB-4**	**5.3 ± 2.8**	**7.0 ± 4.0**	**3.50 × 10^−2^**	
post-	ALT	37.1 ± 81.6	25.5 ± 15.5	4.84 × 10^−1^	
	T.Bil	0.9 ± 0.4	0.9 ± 0.3	7.92 × 10^−1^	
	Alb	3.8 ± 0.4	3.8 ± 0.4	9.71 × 10^−1^	
	PT	85.9 ± 19.2	82.2 ± 14.8	3.20 × 10^−1^	
	PLT	14.0 ± 6.6	11.9 ± 5.4	1.93 × 10^−1^	
	**AFP**	**7.5 ± 6.1**	**14.5 ± 18.0**	**3.52 × 10^−2^**	
	M2BPGi	2.7 ± 2.2	3.5 ± 2.8	1.65 × 10^−1^	

**Table 4 biomedicines-07-00087-t004:** Summary of the equation and prediction score for recurring HCC in liver cirrhosis and non-cirrhosis.

#miRNAs	Equation	ACC (%)	REC (%)	PRE (%)	SPE (%)	LOOCV (%)	ACCALL (%)
**4**	**y = 1.62 × [miR-211-3p] − 1.43 × [miR-6826-3p] + 0.65 × [miR-1236-3p] − 0.84 × [miR-4448] − 1.19**	**85.3**	**86.1**	**77.8**	**85.3**	**87.0**	**87.0**
3	y = 1.93 × [miR-211-3p] − 2.16 × [miR-6826-3p] + 0.80 × [miR-1236-3p] − 5.48	82.9	77.8	76.2		85.5	88.4
2	y = 1.84 × [miR-211-3p] − 1.35 × [miR-6826-3p] − 5.07	79.0	72.8	71.2		81.2	81.2

Abbreviations: Bold indicates Equation (6).

**Table 5 biomedicines-07-00087-t005:** DAA treatment in HCC-naive patients (exosome study).

Summary of Analysis Groups					
Presence or Absence HCC Occurrence	No.	Age	Sex	Liver Disease	DAA Treatment
No	55	66.2 ± 10.6	F:29/M:26	non-cirrhosis: 8 /liver cirrhosis: 47	AD:32/SL:20/SR:3
Yes	15	71.8 ± 8.0	F:8/M:7	non-cirrhosis: 10 /liver cirrhosis: 5	AD:5/SL:8/SR:2
**Clinical Data by Analysis Group**					
**Treatment**	**Items**	**No Occurrence**	**Occurrence**	***p*-Value**	
pre-	ALT	57.1 ± 31.9	54.5 ± 25.2	7.73 × 10^−1^	
	T.Bil	0.9 ± 0.4	0.7 ± 0.3	8.08 × 10^−2^	
	Alb	3.6 ± 0.6	3.8 ± 0.5	2.88 × 10^−1^	
	PT	82.1 ± 26.6	81.2 ± 26.8	8.63 × 10^−1^	
	PLT	10.9 ± 5.5	12.9 ± 6.3	2.36 × 10^−1^	
	AFP	34.2 ± 63.3	18.6 ± 19.2	3.54 × 10^−1^	
	**M2BPGi**	**6.0 ± 3.3**	**3.9 ± 2.0**	**1.84 × 10^−2^**	
	FIB-4	6.5 ± 3.9	5.4 ± 3.4	3.13 × 10^−1^	
post-	ALT	29.7 ± 26.1	23.3 ± 11.9	3.61 × 10^−1^	
	T.Bil	0.9 ± 0.5	0.8 ± 0.3	4.62 × 10^−1^	
	Alb	4.5 ± 5.0	4.0 ± 0.5	6.87 × 10^−1^	
	PT	81.2 ± 25.8	88.5 ± 33.8	1.18 × 10^−1^	
	PLT	11.7 ± 5.7	14.2 ± 6.0	1.51 × 10^−1^	
	**AFP**	**7.8 ± 6.3**	**25.6 ± 49.1**	**1.64 × 10^−2^**	
	M2BPGi	3.4 ± 2.3	2.37 ± 1.7	1.50 × 10^−1^	

**Table 6 biomedicines-07-00087-t006:** Summary of the prediction score for HCC occurrence.

#miRNAs	Chosen mRNAs	ACC (%)	REC (%)	PRE (%)	SPE (%)	LOOCV (%)	ACCALL (%)
4	miR-762, miR-8069, miR-7847-3p, miR-7846-3p	85.5	74.5	66.2		87.1	91.4
3	miR-762, miR-8069, miR-6090	83.4	68.9	64.0		87.1	91.4
2	miR-762, miR-8069	83.3	71.4	60.6	87.1	85.7	85.7

**Table 7 biomedicines-07-00087-t007:** DAA treatment in HCC-naive patients (liver study).

Summary of Analysis Groups
Presence or Absence of HCC Occurrence	No.	Age	Sex	Liver Disease	DAA Treatment
No	36	62.3 ± 5.9	F:23/M:13	F0/F1/F2/F31/22/7/6	AD:23/SL:5/SR:3/VIK:5
Yes	7	62.7 ± 5.3	F:2/M:5	F0/F1/F2/F30/3/3/1	AD:5/SL:1/SR:1
**Clinical Data by Analysis Group**
	**Items**	**SVR-non-HCC**	**SVR-HCC**	***p*-Value**	
pre-treatment	fibrosis stage	1.5 ± 0.8	1.7 ± 0.8	5.22 × 10^−1^	
**ALT**	**48.8 ± 33.5**	**93.1 ± 50.4**	5.30 × 10^−3^	
T.BIL	0.7 ± 0.3	0.9 ± 0.3	1.57 × 10^−1^	
**PLT**	**18.5 ± 5.0**	**12.8 ± 4.2**	6.97 × 10^−3^	
**ALB**	**4.3 ± 0.3**	**4.0 ± 0.2**	1.02 × 10^−2^	
**AFP**	**5.8 ± 7.7**	**17.6 ± 23.6**	1.84 × 10^−2^

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
