# Peer review of "Circulating Exosomal miRNA Profiles Predict the Occurrence and Recurrence of Hepatocellular Carcinoma in Patients with Direct-Acting Antiviral-Induced Sustained Viral Response"

_biomedicines, 2019, doi:10.3390/biomedicines7040087_

Round 1
Reviewer 1 Report
In the present study, the authors demonstrate that a specific set of miRNAs, three or four miRNAs, could be a prognostic factor for assessing the occurrence and recurrence of HCC in HCV-infected HCC patients after DAA therapy. Although most HCV patients achieve sustained viral response (SVR), there is a still high risk of developing HCC occurrence/recurrence after HCV elimination. Hence, it is important to investigate the expression profiles between SVR-HCC group and SVR-non-HCC group to find critical factor relevant to HCC development. Although authors present four miRNAs (miR-4718, miR-642a-5p, miR-6826-3p, miR-762) as potential miRNAs which are predictive for HCC occurrence/recurrence after acquiring SVR, the criteria for miRNA selection are not reasonable and overall data are insufficient to support their conclusion. Especially, microarray data of miR-642a-5p and miR-762 were not in Supplementary Table 3-5. The data are not clearly presented and explanation is not enough. Even there is no explanation for the equation. There are multiple major questions which should be clarified. In addition, this manuscript is a much needed correction in English
1) In figure 1. Although authors described figure1A, it is not easy to understand it. Even there is no explanation for figure 1B-D. Detailed explanation is necessary. In addition, figure legend for figure 1D seems to be irrelevant.
2) In microarray analysis, authors used only patients-derived serum samples, not liver tissues. Given that circulating exosomes in blood are released from many organs including liver, it is not sure that these exosomes reflect liver pathology only. Hence, microarray data obtained from human serum samples should be validated in microarray for liver tissues. In addition, healthy control group should be added in microarray analysis for better comparison of microRNA profiles.
Authors evaluated the expressions of four miRNAs (miR-4718, miR-642a-5p, miR-6825-3p, miR-762) in exosomes and liver tissues of human patients in Figure 5. However, their expressional changes are not significant and mostly unchanged. Although only miR-4718 significantly decreased in the R (+) group compared to the R (-) group, it was a slight change. These results suggest that these four miRNAs are unsuitable as predictive miRNA for HCC. In addition, miRNAs isolated from exosome obtained from serum show the different expressional pattern compared with them from liver, supporting that these exosomes rarely reflect liver physiology.
3) Authors classified the HCV-infected patients into 4 groups, and selected several miRNAs from each group. However, there is no explanation how to choose these miRNAs. It is necessary to provide the reasonable explanation for miRNA selection and experimental evidence supporting author’s explanation.
4) In Result 2.1, authors described that ‘We assessed liver fibrosis using mac-2 binding protein glycosylation isomer (M2BPGi), a novel marker of the progression of liver fibrosis in patients with chronic HCV infection’. Although M2BPGi is a novel marker of liver fibrosis in HCV-infected patients, only one marker is not enough to evaluate liver fibrosis. It is necessary to examine the levels of additional markers for liver fibrosis.
5) In Figure 4 and Result 2.5, authors described that that miR-4718, miR-642a-5p, miR-762 and miR-8069 inhibited cell proliferation of Huh 7.5 cells, and miR-4718, miR-6551a-5p, miR-642a-5p, miR-4448, miR-6826-3p, miR-1236-3p and miR-8069 inhibited cell proliferation of HepG2 cells. However, graphs present that the levels of other miRNAs, such as miR-6511a and miR-211, were much lower than that of other miRNAs which authors mentioned. This patterns is also observed in proliferation graphs of HepG2. These inconsistent interpretations of data make this manuscript unreliable and confusing. Authors need to conduct statistical analysis again and describe the data correctly.
6) In this manuscript, many tables are presented to summarize the clinical background of patients as well as the predictive scores for assessing a set of predictive miRNAs for HCC. Tables are useful for organizing large and/or complex data, and is commonly used in presenting the data at a glance. Therefore, table is helpful to understand the huge data. However, the tables in the manuscript is too difficult to be interpreted and rather hinders interpretation. Authors need to reorganize and redisplay them.
7) In Figure 5, four miRNAs which were selected from microarray analysis were not changed significantly between the R(-) and the R(+) group, indicating that microarray data were invalid, although it was not known how to choose these miRNAs (see comment #3). In addition, authors provided simple diagram depicting the potential targets of four miRNAs. The putative targets of miRNA obtained from TargetScan database should be validated by luciferase assay.
8) Please add the detailed explanation and references for the equation. If authors make the equation, it should first be validated.
9) There is no healthy control in all tables, and it is not easy to find out whether the difference between groups is meaningful. Please add the data of healthy control group in tables.
10) There is description irrelevant to the study in the last paragraph of the introduction section. Authors seem to copy & past ‘Author guideline’. Authors need to work hard on writing the manuscript.
11) There is no description about analysis of Figure 5 in figure legends and method section.
12) Authors isolated exosomes from serum, not liver. Hence, it had better change the titles form exosomal miRNAs ~~~~ to circulating exosomal miRNAs~~~”.
Author Response
In the present study, the authors demonstrate that a specific set of miRNAs, three or four miRNAs, could be a prognostic factor for assessing the occurrence and recurrence of HCC in HCV-infected HCC patients after DAA therapy. Although most HCV patients achieve sustained viral response (SVR), there is a still high risk of developing HCC occurrence/recurrence after HCV elimination. Hence, it is important to investigate the expression profiles between SVR-HCC group and SVR non-HCC group to find critical factor relevant to HCC development. Although authors present four miRNAs (miR-4718, miR-642a-5p, miR-6826-3p, miR-762) as potential miRNAs which are predictive for HCC occurrence/recurrence after acquiring SVR, the criteria for miRNA selection are not reasonable and overall data are insufficient to support their conclusion. Especially, microarray data of miR-642a-5p and miR-762 were not in Supplementary Table 3-5. The data are not clearly presented and explanation is not enough. Even there is no explanation for the equation. There are multiple major questions which should be clarified. In addition, this manuscript is a much needed correction in English.
1) In figure 1. Although authors described figure1A, it is not easy to understand it. Even there is no explanation for figure 1B-D. Detailed explanation is necessary. In addition, figure legend for figure 1D seems to be irrelevant.
>We added detailed description of figure 1 and figure 1D has been removed based on reviewer’s comments..
2) In microarray analysis, authors used only patients derived serum samples, not liver tissues. Given that circulating exosomes in blood are released from many organs including liver, it is not sure that these exosomes reflect liver pathology only. Hence, microarray data obtained from human serum samples should be validated in microarray for liver tissues. In addition, healthy control group should be added in microarray analysis for better comparison of microRNA profiles. Authors evaluated the expressions of four miRNAs (miR-4718, miR-642a-5p, miR-6825-3p, miR-762) in exosomes and liver tissues of human patients in Figure 5. However, their expressional changes are not significant and mostly unchanged. Although only miR-4718 significantly decreased in the R (+) group compared to the R (-) group, it was a slight change. These results suggest that these four miRNAs are unsuitable as predictive miRNA for HCC. In addition, miRNAs isolated from exosome obtained from serum show the different expressional pattern compared with them from liver, supporting that these exosomes rarely reflect liver physiology.
> As the reviewer pointed out, there are some microRNAs that are used in the separation, where the difference in expression between carcinogenic and non-carcinogenic cases is not very large. The microRNAs selected in this study are not the ones with the greatest difference in expression depending on the presence or absence of carcinogenesis, but for the most accurate diagnosis of carcinogenesis.
In order to elucidate the significance of microRNAs selected from this perspective in relation to carcinogenesis, microRNA expression analysis in the liver and serum was performed on a microarray. Certainly, as the reviewer points out, circulating exosomes in the blood are released from many organs including the liver. Figure 5 shows how the microRNAs used for classification are expressed in liver tissue of the same disease, the known physiological activity of microRNAs, and the relationship of known target genes of microRNAs. Detailed mechanism analysis will be a subject for further study, but we believe that the microRNAs used for classification can indicate the possibility of being involved in carcinogenesis.
3) Authors classified the HCV-infected patients into 4 groups, and selected several miRNAs from each group. However, there is no explanation how to choose these miRNAs. It is necessary to provide the reasonable explanation for miRNA selection and experimental evidence supporting author’s explanation.
> The selection procedure of miRNAs was described in Section 2.5. In order to clarify it, we replaced “using the linear SVM-based selection procedure” in the third paragraph of Section 3.1 with “using the linear SVM-based selection procedure described in Section 2.5”. The effectiveness of these selected miRNAs was demonstrated in cross-validation tests and permutation tests as discussed in Sections 3.1-3.3.
4) In Result 2.1, authors described that ‘We assessed liver fibrosis using mac-2 binding protein glycosylation isomer (M2BPGi), a novel marker of the progression of liver fibrosis in patients with chronic HCV infection’. Although M2BPGi is a novel marker of liver fibrosis in HCV-infected patients, only one marker is not enough to evaluate liver fibrosis. It is necessary to examine the levels of additional markers for liver fibrosis.
>According to the reviewer’s recommendation, we added FIB-4 index as alternative liver fibrosis marker.
5) In Figure 4 and Result 2.5, authors described that that miR-4718, miR-642a-5p, miR-762 and miR-8069 inhibited cell proliferation of Huh 7.5 cells, and miR-4718, miR-6551a-5p, miR-642a-5p, miR-4448, miR-6826-3p, miR-1236-3p and miR-8069 inhibited cell proliferation of HepG2 cells. However, graphs present that the levels of other miRNAs, such as miR-6511a and miR-211, were much lower than that of other miRNAs which authors mentioned. This patterns is also observed in proliferation graphs of HepG2. These inconsistent interpretations of data make this manuscript unreliable and confusing. Authors need to conduct statistical analysis again and describe the data correctly.
> As the Reviewer pointed out, the miRNA expression experiment for cell proliferation and apoptosis results and the contribution by aberrant expression of these miRNAs on carcinogenesis were difficult to understand, so we reconfirmed date and revised result 3.5, discussion, and figure 4 and 5.
6) In this manuscript, many tables are presented to summarize the clinical background of patients as well as the predictive scores for assessing a set of predictive miRNAs for HCC. Tables are useful for organizing large and/or complex data, and is commonly used in presenting the data at a glance. Therefore, table is helpful to understand the huge data. However, the tables in the manuscript is too difficult to be interpreted and rather hinders interpretation. Authors need to reorganize and redisplay them.
> In line with your comments, table 1, 3, and 5 are divided into groups and individual information, respectively. In addition, according to the comments from Reviewer 2, ROC curves and heat maps were newly added (Supplemental Figures 1 and 2) to visualize data and some of computational results.
7) In Figure 5, four miRNAs which were selected from microarray analysis were not changed significantly between the R(-) and the R(+) group, indicating that microarray data were invalid, although it was not known how to choose these miRNAs (see comment #3). In addition, authors provided simple diagram depicting the potential targets of four miRNAs. The putative targets of miRNA obtained from TargetScan database should be validated by luciferase assay.
>The microRNA selection method was described in the response of comment # 3. In accordance with the reviewer's suggestion, the in vitro experiment results reassessed as described in the rebuttal of section 5. Since miR-4718 is only involved in cell proliferation under certain conditions, The relationship between the target gene (HDAC8) and miR-4718 was cited in previous report in the discussion section and removed from Figure 5.
8) Please add the detailed explanation and references for the equation. If authors make the equation, it should first be validated.
> Equations were derived as the results of the learning phase of Support Vector Machine (SVM). SVM has been widely used for various medical prediction problems. Equations were verified by means of cross validation tests and permutation tests, where details are given in Sections 3.1-3.3. We have added more details on SVM in the beginning of Section 2.5 with citing a review article by Huang et al. (2018). We have also added a sentence just below Equation (1).
9) There is no healthy control in all tables, and it is not easy to find out whether the difference between groups is meaningful. Please add the data of healthy control group in tables.
>As pointed out by reviewer, having the target gene information is more useful for understanding the mechanism. However, the results of our in vitro experiments indicate that the approximate expression pattern may be related to carcinogenesis. It is also difficult to perform additional experiments because the time allowed for review responses is too short.
10) There is description irrelevant to the study in the last paragraph of the introduction section. Authors seem to copy & past ‘Author guideline’. Authors need to work hard on writing the manuscript.
>Thank you for your comment, we excluded unnecessary items that are not directly related to the paper.
11) There is no description about analysis of Figure 5 in figure legends and method section.
>Thank you for your comment, we explained legend of figure 5 in detail and method section.
12) Authors isolated exosomes from serum, not liver. Hence, it had better change the titles form exosomal miRNAs ~~~~ to circulating exosomal miRNAs~~~”.
>We change the title as reviewer’s recommendation.
Reviewer 2 Report
The paper entitled “Exosomal miRNA profiles predict the occurrence and recurrence of hepatocellular carcinoma in patients with direct-acting antiviral–induced sustained viral response” present important information about miRNAs cargo in circulating vesicles/complexes. The generating panel of miRNAs could be of interest for further prediction of HCC recurrence. However, some clarifications are necessary for further improvement of the manuscript.
The term exosomes is a very concrete kind of vesicles. The paper, however, is based on the trap of circulating miRNAs using exoquick. This reagent enrich the sample in vesicles, but also miRNA protein complexes. To claim for miRNA associate to vesicles, further test would be necessary (I suggest the read of two articles; (MISEV2018): a position statement of the International Society for Extracellular Vesicles and update of the MISEV2014 guidelines, Journal of Extracellular Vesicles, 7:1, DOI: 10.1080/20013078.2018.1535750, and Arroyo, J. D., Chevillet, J. R., Kroh, E. M., Ruf, I. K., Pritchard, C. C., Gibson, D. F., … Tewari, M. (2011). Argonaute2 complexes carry a population of circulating microRNAs independent of vesicles in human plasma. Proceedings of the National Academy of Sciences of the United States of America, 108(12), 5003–5008. https://doi.org/10.1073/pnas.1019055108. Since the exact origin of miRNA not seems to be the point of the article, I suggest to replace the “exosomal miRNA” for “circulating miRNA”, and mention in the introduction/discussion that was purified by exoquick Most papers that try to generate a miRNA panel, works with the concepts sensitivity and specificity, and generate roc curves… why did you prefer the concepts Precision, Recall and Accuracy? In some predictions (i.e. Table 2) the model with 3 miRNAs seems to be more accurate than the the model with 4 miRNAs. Is this possible? Figure 3, upper panel, why the expression values for 7847-3p changes between 4 and 3 miRNAs? Actually the 3 panels should present same data (as in figure 2) but just change the scale of the y axis right? If that is the case, perhaps a single representation would be more accurate. Additionally, a heatmap with supervised clustering (recurrence vs non recurrence?) could be presented with all miRNAs assayed in the array. Could you clarify if the analysis of miRNAs in liver and exosomes was performed in matching patients, and the conclusions of a same pattern was drawn from those match (i.e. the same patient suffer a decrease in the liver and in circulating miRNAs) I miss a final summary of the proposed miRNA panel for each prediction. I will suggest present a a roc curve for each panel. Pag 10 line 268… looks like a paragraph from some draft version? I did not completely understand the comparison between liver and circulating miRNAs. Among the miRNAs that are valuable as predictor, you just choose those which change in the same direction in exosomes and liver. However, opposite tendencies between EVs and liver had been observed (miR122 is a notorious example). Actually, the 4 that you analyzed seems to be not much different either circulating or in the liver. You finally proposed some genes that may be related to the miRNAs that you selected. How is the expression of those genes in the liver? If you could not check yourselves, you certainly can access expression databases to see how is the expression of those genes in HCC. In the discussion, it can be compared the miRNAs selected in this work with previous panels of circulating miRNAs found in other studies for HCC recurrence.
Author Response
The paper entitled “Exosomal miRNA profiles predict the occurrence and recurrence of hepatocellular carcinoma in patients with direct-acting antiviral–induced sustained viral response” present important information about miRNAs cargo in circulating vesicles/complexes. The generating panel of miRNAs could be of interest for further prediction of HCC recurrence. However, some clarifications are necessary for further improvement of the manuscript.
(1) The term exosomes is a very concrete kind of vesicles. The paper, however, is based on the trap of circulating miRNAs using exoquick. This reagent enrich the sample in vesicles, but also miRNA protein complexes. To claim for miRNA associate to vesicles, further test would be necessary (I suggest the read of two articles; (MISEV2018): a position statement of the International Society for Extracellular Vesicles and update of the MISEV2014 guidelines, Journal of Extracellular Vesicles, 7:1, DOI: 10.1080/20013078.2018.1535750, and Arroyo, J. D., Chevillet, J. R., Kroh, E. M., Ruf, I. K., Pritchard, C. C., Gibson, D. F., … Tewari, M. (2011). Argonaute2 complexes carry a population of circulating microRNAs independent of vesicles in human plasma. Proceedings of the National Academy of Sciences of the United States of America, 108(12), 5003–5008. https://doi.org/10.1073/pnas.1019055108. Since the exact origin of miRNA not seems to be the point of the article, I suggest to replace the “exosomal miRNA” for “circulating miRNA”, and mention in the introduction/discussion that was purified by exoquick.
>According to reviewer’s suggestion, we argue that miRNA in circulating blood was analyzed in an environment similar to exosome, which was generated by exoquick. The above content was added to the introduction, and after that circulating miRNA purified by exoquick was described as exosomal miRNA and presented two articles were added in references.
(2) Most papers that try to generate a miRNA panel, works with the concepts sensitivity and specificity, and generate roc curves… why did you prefer the concepts Precision, Recall and Accuracy?
> Sensitivity is equivalent to recall. In addition to precision, recall, and accuracy, we have newly added specificity values. Since the number of positive data was very small and independent test data was not available, it was quite difficult to obtain meaningful ROC curves. Therefore, we employed cross validation test. However, ROC curves might be useful for visualization of the learning results (not prediction results) even in such cases. Therefore, we have added ROC curves (Supplemental Figure 1) where all samples were included in both the training and test datasets (otherwise, quite a small number of positive samples would be available). We have also added some sentences in the latter part of Section 2.5.
(3) In some predictions (i.e. Table 2) the model with 3 miRNAs seems to be more accurate than the model with 4 miRNAs. Is this possible?
> Yes. It is possible because SVM does not directly try to maximize the accuracy (it minimizes the weighted sum of the square of inverse of the margin and the classification error).
(4) Figure 3, upper panel, why the expression values for 7847-3p changes between 4 and 3 miRNAs?
> Thank you for pointing out our mistake. We used different miRNAs between 4 and 3 miRNAs. We have replaced Figure 3 with a correct one.
(5) Actually the 3 panels should present same data (as in figure 2) but just change the scale of the y axis right?
> In Figure 2, it is so. However, (somewhat) different miRNAs were used in Figure 3, as mentioned in the response to Comment (4).
(6) If that is the case, perhaps a single representation would be more accurate. Additionally, a heatmap with supervised clustering (recurrence vs non recurrence?) could be presented with all miRNAs assayed in the array.
> Since different miRNAs were used in Figure 3 and we would like to use the same format for all three cases, we did not change the formats of Figures 2 and 3.
According to the suggestion, we have added heat maps with supervised clustering (e.g., recurrence vs. non-recurrence) as Supplemental Figure 2, where 50 relevant miRNAs (selected by the procedure in Section 2.5) were used in each case for the sake of clearer visualization. We have also added a sentence on it in the latter part of Section 2.5. However, it is difficult to see clear tendencies discriminating positive and negative samples from the heat maps. Therefore, we think that Figures 2 and 3 are still useful.
(7) Could you clarify if the analysis of miRNAs in liver and exosomes was performed in matching patients, and the conclusions of a same pattern was drawn from those match (i.e. the same patient suffer a decrease in the liver and in circulating miRNAs) I miss a final summary of the proposed miRNA panel for each prediction.
> Since there are only a few cases with both serum and liver samples, it is difficult to perform an analysis on this dataset. We would like to consider adding more samples in the future.
(8) I will suggest present a a roc curve for each panel. Pag 10 line 268… looks like a paragraph from some draft version?
> As mentioned in the response to Comment (2), we have added ROC curves as Supplemental Figure 1. Low accuracy results of the permutation test (compared with cross validation accuracies) suggest that the possibility of overfitting is low.
(9) I did not completely understand the comparison between liver and circulating miRNAs. Among the miRNAs that are valuable as predictor, you just choose those which change in the same direction in exosomes and liver. However, opposite tendencies between EVs and liver had been observed (miR122 is a notorious example). Actually, the 4 that you analyzed seems to be not much different either circulating or in the liver. You finally proposed some genes that may be related to the miRNAs that you selected. How is the expression of those genes in the liver? If you could not check yourselves, you certainly can access expression databases to see how is the expression of those genes in HCC. In the discussion, it can be compared the miRNAs selected in this work with previous panels of circulating miRNAs found in other studies for HCC recurrence.
>As the reviewer pointed out, the difference in the expression of microRNA selected this time between carcinogenic and non-carcinogenic cases is not very large. In order to clarify the significance of microRNA used for carcinogenesis prediction, we tried to compare circulating microRNA with microRNA in liver tissue. Figure 5 summarizes the relationship between microRNAs, target genes, and biological functions. However, hsa-miR-6826-3p and hsa-miR-762, which were used to predict carcinogenesis at the present time, were searched in databases, etc., but information on target genes and functions could not be obtained.
Round 2
Reviewer 1 Report
Authors answered most of reviewer's comments and have improtved the quality and importance of the manuscirpt. Thus, I do not have further comments.